# Association of digoxin with mortality in patients with advanced chronic kidney disease: A population-based cohort study

Lii-Jia Yang[1,2], Shan-Min Hsu[2], Ping-Hsun Wu[2,3], Ming-Yen Lin[2,4,5], Teng-Hui Huang[2], Yi-Ting Lin[3,6,7], Hung-Tien Kuo[2,4], Yi-Wen Chiu[2,4], Shang-Jyh Hwang[2,4], Jer-Chia Tsai[2,4]*, Hung-Chun Chen[2,4]

1 Department of Internal Medicine, Kaohsiung Municipal CiJin Hospital, Kaohsiung, Taiwan, 2 Division of Nephrology, Department of Internal Medicine, Kaohsiung Medical University Hospital, Kaohsiung, Taiwan, 3 Institute of Clinical Medicine, College of Medicine, Kaohsiung Medical University, Kaohsiung, Taiwan, 4 Department of Renal Care, College of Medicine, Kaohsiung Medical University, Kaohsiung, Taiwan, 5 Master of Public Health Degree Program, College of Public Health, National Taiwan University, Taipei, Taiwan, 6 Department of Family Medicine, Kaohsiung Medical University Hospital, Kaohsiung, Taiwan, 7 Department of Family Medicine, Kaohsiung Municipal Hsiao–Kang Hospital, Kaohsiung, Taiwan

* jerchia.tsai@gmail.com

**Data Availability Statement:** The data used in our study were limited to research purposes only and cannot be made publicly available under regulation of the Personal Information Protection Act in

## Abstract

Digoxin is commonly prescribed for heart failure and atrial fibrillation, but there is limited data on its safety in patients with chronic kidney disease (CKD). We conducted a population-based cohort study using the pre-end stage renal disease (ESRD) care program registry and the National Health Insurance Research Database in Taiwan. Of advanced CKD patient cohort (N = 31,933), we identified the digoxin user group (N = 400) matched with age and sex non-user group (N = 2,220). Multivariable Cox proportional hazards and sub-distribution hazards models were used to evaluate the association between digoxin use and the risk of death, cardiovascular events (acute coronary syndrome, ischemic stroke, or hemorrhagic stroke) and renal outcomes (ESRD, rapid decline in estimated glomerular filtration rate—eGFR, or acute kidney injury). Results showed that all-cause mortality was higher in the digoxin user group than in the non-user group, after adjusting for covariates (adjusted hazard ratio, aHR 1.63; 95% CI 1.23–2.17). The risk for acute coronary syndrome (sub-distribution hazard ratio, sHR 1.18; 95% CI 0.75–1.86), ischemic stroke (sHR 1.42; 95% CI 0.85–2.37), and rapid eGFR decline (sHR 1.00 95% CI 0.78–1.27) was not significantly different between two groups. In conclusion, our study demonstrated that digoxin use was associated with increased mortality, but not cardiovascular events or renal function decline in advanced CKD patients. This finding warns the safety of prescribing digoxin in this population. Future prospective studies are needed to overcome the limitations of cohort study design.

## Introduction

Digoxin, a cardiac glycoside, decreases heart rate and increases myocardial contractility by inhibiting cellular sodium-potassium adenosine triphosphatase (N+/K+-ATPase). Digoxin has been prescribed to treat heart failure (HF) or atrial fibrillation (AF).

Taiwan. The raw data were obtained from the following sources and can be made available to qualified researchers upon request: NHIRD datasets H_NHI_OPDTE, H_NHI_IPDTE, H_NHI_DRUGE, H_NHI_OPDTO, H_NHI_IPDTO, H_NHI_DRUGO, H_NHI_ENROL, H_NHI_CATAS, and H_OST_DEATH from the Health and Welfare Data Science Center, Department of Statistics, Ministry of Health and Welfare, Taiwan (https://dep. mohw.gov.tw/DOS/np-2497-113.html). Pre-ESRD care program dataset from the National Health Insurance Administration, Ministry of Health and Welfare (https://www.nhi.gov.tw/Content_List. aspx?n=2D2FAF5214807829&topn= 787128DAD5F71B1A).

**Funding:** This study was supported by the Kaohsiung Municipal CiJin Hospital under Grant Kmch-108-001 to LJY, and partially supported by grants from the Kaohsiung Medical University Hospital (KMUH104-4M07, KMUH104-4M08, KMUH104-4R10, KMUH105-5R18, KMUH106-6R18, and KMUH107-7R17) and Ministry of Science and Technology, Taiwan, R.O.C. (MOST104-2511-S-037-004-MY2, MOST106-2511-S-037-002, and MOST107-2511-H-037-006-MY2) to JCT. The funders had no role in the study design, data collection and analysis, decision to publish, or manuscript preparation.

**Competing interests:** Co-author Ping-Hsun Wu is a fellow of PLOS ONE Editorial Board Members. This does not alter our adherence to PLOS ONE policies on sharing data and materials.

In high-risk subgroups of patients with HF, such as those with New York Heart Association (NYHA) class III–IV, left ventricular ejection fraction (LVEF) <25% and cardiothoracic ratio >55%, digoxin was associated with a lower risk of all-cause mortality or hospitalization [1, 2]. Current guidelines recommend digoxin be considered for symptomatic HF patients to reduce hospitalization risk, despite receiving standard therapy, including beta-blockers, angiotensin-converting enzyme inhibitors, angiotensin-receptor blockers, or mineralocorticoid receptor antagonists [3].

By contrast, in patients with AF, treatment with digoxin could be associated with increased mortality [4, 5]. However, in a recent meta-analysis of randomized control trials, the clinical effects of digoxin on all-cause mortality, serious adverse events, quality of life, heart failure, and stroke in patients with AF remains unclear [6]. Current guidelines recommend digoxin as a rate control agent in patients with AF, particularly in those with concomitant HF [7, 8].

Patients with chronic kidney disease (CKD) have multiple comorbidities, including HF and AF [9, 10], making CKD patients possible candidates for using digoxin. However, digoxin is predominantly excreted by the kidneys, so impaired renal function can significantly influence its pharmacokinetics [11]. In addition, digoxin has a narrow therapeutic-toxicity range [12], possesses multiple drug-drug interactions [13], and the manifestation of its toxicity, nausea and vomiting, could mimic the uremic symptoms of late CKD. Notably, it was reported that CKD did not directly affect all-cause and cardiovascular mortality in patients with AF taking digoxin [14]. However, there is still a concern in prescribing digoxin for CKD patients because of its safety. Currently, limited data is addressing its safety in CKD patients from the population-based approach. This study aimed to investigate the effect of digoxin on all-cause mortality, cardiovascular events, and renal outcomes in a nationwide CKD cohort in Taiwan.

## Materials and methods

### A brief overview of Taiwan pre-end-stage renal disease (ESRD) pay-for-performance program

National Health Insurance (NHI) is a mandatory, universal, and single-payer insurance system in Taiwan, covering over 99% of the population. The Pre- ESRD care program, implemented by NHI in 2006, is a pay-for-performance healthcare model designed to prevent or delay dialysis, avoid uremic complications, and reduce health care costs through patient-centered case management by a multidisciplinary team. Eligibility criteria were individuals with CKD stages 3b-5 (eGFR<45 mL/min/1.73 m$^2$), or those with proteinuria (urine protein to creatinine ratio, UPCR >1000 mg/g). Participating patients were required to attend a hospital at least quarterly for clinical and laboratory evaluation by a nephrologist, CKD education provided by a renal nurse, and a diet consultation by a dietitian. Participating health care providers received additional payment for patient enrollment and each follow-up visit, and they were also rewarded if they achieved predetermined targets, such as a reduced estimated glomerular filtration rate (eGFR) progression (<6 ml/min/1.73m$^2$), complete remission of proteinuria (UPCR <200 mg/g), and vascular access before dialysis.

### Study population and cohort

We conducted a retrospective cohort study using the Pre-ESRD care program registry linked with the National Health Insurance Research Database (NHIRD), containing detailed information on inpatient and outpatient services. To protect patients' privacy, NHIRD had made all data fully anonymized by replacing all personal identification with surrogate numbers before researchers accessed them and further analyzed them. We included patients diagnosed with

CKD, based on the International Classification of Diseases, Ninth Revision, Clinical Modification (ICD-9-CM) codes 585 and 581.9, on at least two outpatient visits or one hospitalization between January 1, 2007 and December 31, 2011. Individuals younger than 18 years of age, with early-stage CKD (stages 1-3a), with eGFR recorded <3 times, or without baseline laboratory data were excluded. Individuals who died within three months of enrollment, underwent dialysis within three months of being enrolled, or received a renal transplant were also excluded. The index date was the date the patients were enrolled in the pre-ESRD program. CKD patients receiving digoxin treatment were defined as digoxin users, then each user was matched with five untreated control patients selected from the same registry, according to age and sex. Patients were followed until death, ESRD, or until 2012, whichever occurred first. This study was approved by the Institutional Review Board (IRB) of Kaohsiung Medical University Hospital (KMUHIRB-EXEMPT(I)-20180035), and the requirement for informed consent was waived.

## Measurement of outcomes

Outcomes were the occurrence of all-cause mortality, major cardiovascular events (composite endpoints of acute coronary syndrome [ACS], ischemic stroke and hemorrhagic stroke), and renal outcomes (ESRD, rapid eGFR decline and acute kidney injury—AKI) during the study period. Death was ascertained based on the evidence of patient withdrawal from the NHI claim database. ACS, ischemic stroke, and hemorrhagic stroke were defined as hospitalization for these vascular events, which were validated in previous studies [15, 16]. For example, ICD-9-CM codes 433 (occlusion of cerebral arteries) and 434 (stenosis of precerebral arteries) were used to extract from NHIRD study subjects with ischemic stroke and were admitted for the specific diagnosis.

Patients with a subsequent diagnosis of ESRD were identified from the Registry for Catastrophic Illness Patient Database. The accuracy of ESRD diagnosis was ascertained because all ESRD patients in Taiwan were reviewed and issued a catastrophic illness registration card from the NHI Administration for waiving the co-payments for long-term dialysis. Moreover, the rapid eGFR decline was defined as a one-year eGFR slope >5 mL/min per 1.73 m$^2$ after the index date. AKI events were identified according to ICD-9-CM codes 584 [17].

## Comorbidities and exposure to confounding medications

The following comorbidities were identified as potential confounders: diabetes mellitus (DM), hypertension, hyperlipidemia, coronary artery disease, cerebrovascular disease, AF, HF, gout, and malignancy (S1 Table). The definition of DM, hypertension, and hyperlipidemia required both the specific ICD-9-CM codes and the use of disease-defining medications for a minimum of 90 days. Comorbidities were scored based on a comorbidity index developed for Chinese ESRD patients [18].

We also retrieved details regarding medication usage during the study, including antiplatelet agents/warfarin, antihypertensive drugs (angiotensin-converting enzyme inhibitors, angiotensin receptor blockers, beta-blockers, diuretics, and calcium channel blockers), statins, oral antidiabetic agents, insulin, and nonsteroidal anti-inflammatory drugs (both traditional agents and cyclooxygenase-2 selective inhibitors; S2 Table). Medication use was defined as drugs with an accumulated duration of more than 28 days during the study period.

## Characteristics of clinical data

The abbreviated Modification of Diet in Renal Disease equation was used to calculate the eGFR [19]. Baseline eGFR was calculated from the last recorded serum creatinine level before the index date. The change of eGFR between one year of follow-up and baseline was then

calculated for each subject. Clinical data included body weight, blood pressure, hematocrit, serum albumin, and UPCR. The stage of CKD was defined according to baseline eGFR.

## Statistical analysis

Baseline descriptive data were described as mean ± standard deviation for continuous variables, and frequency and percentage were displayed for categorical variables. The incidence rate ratios (IRRs) and 95% confidence intervals (95% CIs) of outcomes (all-cause mortality, cardiovascular events, and renal outcomes) for digoxin users versus non-users were examined by using the Poisson regression model [20]. Regarding the all-cause mortality outcome, we employed Cox proportional hazards model. Furthermore, we applied the Fine and Gray sub-distribution hazards model to clarify the competing risk of death and the effects of digoxin on the cardiovascular and renal outcomes [21]. We applied the multivariable models to adjust the confounders for urbanization, socioeconomic status, comorbid disorders, clinical characteristics, and each medication class.

Intention-to-treat (ITT) and as-treated (AT) analyses were conducted. For ITT analysis, digoxin users and non-users were followed until the end of the study according to their original treatment allocations regardless of adherence by patients, subsequent withdrawal, or any change in treatment status over time. For AT analysis, the person-time was censored on the day of digoxin discontinuation. Patients were allowed to have a grace period of up to 30 days between prescription dates when calculating continuous therapy. Both approaches are associated with different biases and might complement each other [22]. All analyses were performed using SAS statistical software (version 9.2; SAS Institute Inc., www.sas.com). All statistical tests were two-sided. $P < 0.05$ was considered statistically significant.

## Sensitivity analysis

To assess the robustness of our results, we conducted sensitivity analyses using: (1) a logistic regression model that included age, sex, urbanization, socioeconomic status, comorbidities, clinical characteristics, and concurrent medications as covariates to compute the propensity score, and by performing ITT analysis based on patients matched by propensity score; (2) propensity score-matched Cox regression models and performing AT analysis; (3) re-defined digoxin users as cumulative use ≥28 days and analyzing the original age- and sex-matched cohort with multivariable-adjusted models; (4) re-defined digoxin users as cumulative use ≥56 days; and (5) re-defined digoxin users as cumulative use ≥84 days.

## Results

### Baseline characteristics

As shown in Fig 1, a total of 31,993 patients had advanced CKD diagnosed between 1 January 2007 and 31 December 2011. We identified 440 CKD patients treated with digoxin and 2,200 non-users, by age- and sex-matching process. Table 1 shows the baseline characteristics of the study population. Patients who received digoxin were more likely to have DM, coronary artery disease, cerebrovascular disease, AF, chronic HF, and gout; however, they were less likely to have hypertension. Higher comorbidities scores were found for digoxin users than for non-users. Baseline clinical data demonstrated that digoxin users had higher eGFR and hematocrit, but lower systolic and diastolic blood pressure. A higher proportion of patients in the digoxin user group received concomitant medical treatment, including antiplatelet agents/warfarin, beta-blockers, diuretics, oral antidiabetic agents, and insulin, compared to those who did not use digoxin. The mean follow-up times for digoxin users and non-users were 24.8 and 26.4 months, respectively.

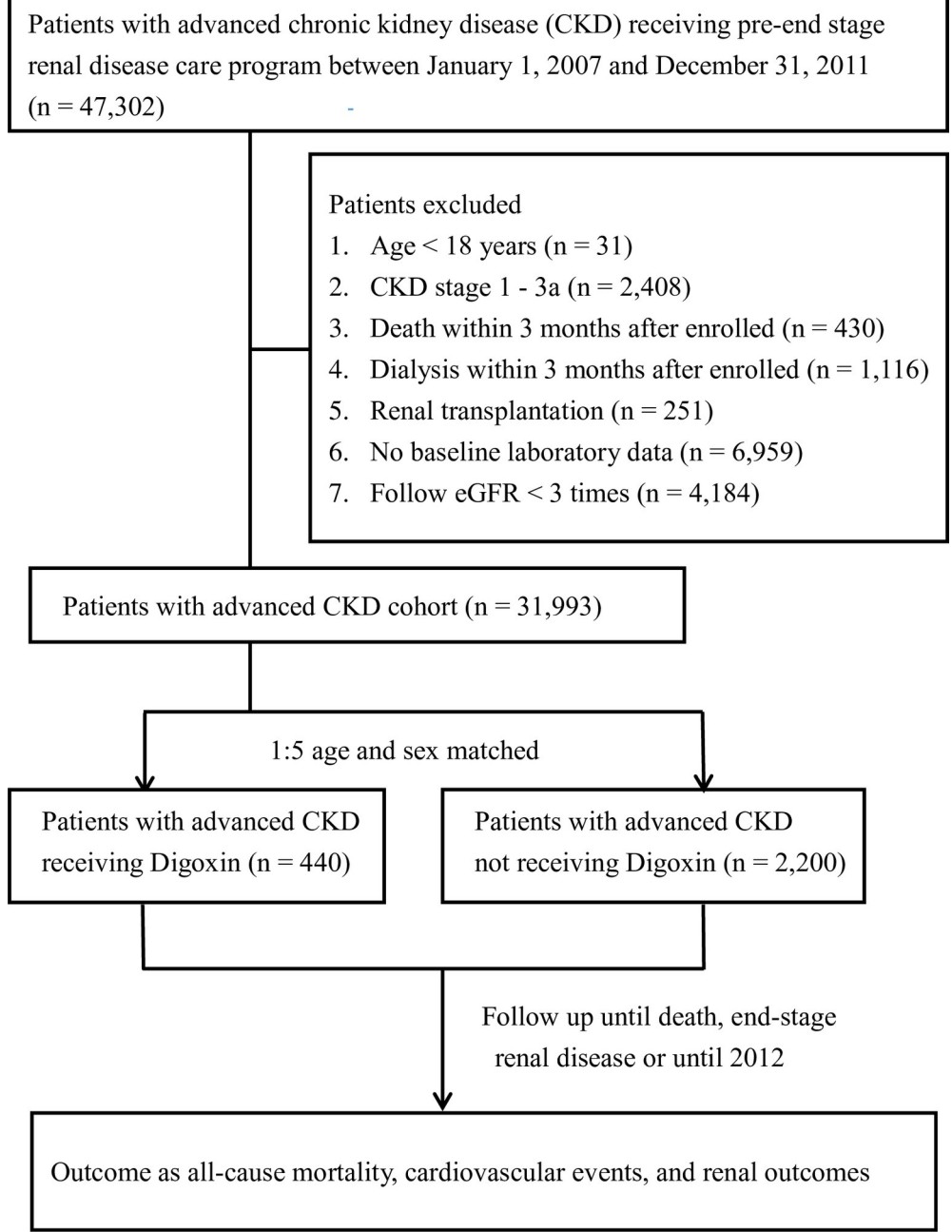

**Fig 1. Study flowchart.**

## All-cause mortality, cardiovascular events, and renal outcomes

From ITT analysis, digoxin-treated CKD patients had a higher risk of all-cause mortality (IRR 2.25, 95% CI 1.81–2.80), ACS (IRR 1.86, 95% CI 1.30–2.67), ischemic stroke (IRR 1.91, 95% CI 1.28–2.86), and AKI (IRR 1.70, 95% CI 1.34–2.16), compared with non-digoxin-treated CKD patients (Table 2).

After adjusting for urbanization, socioeconomic status, comorbid disorders, clinical characteristics, medications, and competing risks of mortality, digoxin use was independently

**Table 1. Baseline characteristics among patients with chronic kidney disease receiving digoxin or not.**

| Characteristics | With Digoxin (n = 440) | Without Digoxin (n = 2,200) | *P* value |
|---|---|---|---|
| Age, yr, mean ± SD | 73.9 ± 9.9 | 73.9 ± 9.9 | >0.999 |
| Age group, yr, n (%) | | | >0.999 |
| 18–39 | 2 (0.5%) | 10 (0.5%) | |
| 40–59 | 47 (10.7%) | 235 (10.7%) | |
| 60–79 | 249 (56.6%) | 1245 (56.6%) | |
| ≥80 | 142 (32.3%) | 710 (32.3%) | |
| Men, n (%) | 288 (65.5%) | 1440 (65.5%) | >0.999 |
| Urbanization level, n (%) | | | 0.525 |
| City area | 308 (70.0%) | 1576 (71.6%) | |
| Rural area | 132 (30.0%) | 624 (28.4%) | |
| Socioeconomic status[a], n (%) | | | 0.743 |
| Low economics | 174 (39.5%) | 834 (37.9%) | |
| Moderate economics | 109 (24.8%) | 579 (26.3%) | |
| High economics | 157 (35.7%) | 787 (35.8%) | |
| CKD stage, n (%) | | | 0.126 |
| 3b | 152 (34.5%) | 685 (31.1%) | |
| 4 | 186 (42.3%) | 907 (41.2%) | |
| 5 | 102 (23.2%) | 608 (27.6%) | |
| Comorbidities[b], n (%) | | | |
| Diabetes mellitus | 254 (57.7%) | 1067 (48.5%) | <0.001 |
| Hypertension | 332 (75.5%) | 1784 (81.1%) | 0.008 |
| Hyperlipidemia | 150 (34.1%) | 671 (30.5%) | 0.153 |
| Coronary artery disease | 203 (46.1%) | 534 (24.3%) | <0.001 |
| Cerebrovascular disease | 72 (16.4%) | 244 (11.1%) | 0.002 |
| Atrial fibrillation | 130 (29.5%) | 46 (2.1%) | <0.001 |
| Heart failure | 227 (51.6%) | 237 (10.8%) | <0.001 |
| Gout | 138 (31.4%) | 572 (26.0%) | 0.024 |
| Malignancy | 35 (8.0%) | 214 (9.7%) | 0.284 |
| Comorbidities score[c], median (IQR) | 7.0 (5.0, 10.0) | 4.0 (2.0, 7.0) | <0.001 |
| Clinical characteristics | | | |
| Body weight, kg | 62.8 ± 12.4 | 63.9 ± 11.8 | 0.072 |
| eGFR, ml/min per 1.73 m$^2$ | 24.8 ± 10.3 | 23.3 ± 10.8 | 0.008 |
| SBP, mmHg | 129.5 ± 19.5 | 135.2 ± 18.1 | <0.001 |
| DBP, mmHg | 72.0 ± 13.1 | 74.0 ± 11.6 | 0.002 |
| Hematocrit, % | 34.0 ± 6.0 | 33.2 ± 5.6 | 0.011 |
| Serum albumin, g/dl | 3.9 ± 0.5 | 4.0 ± 0.5 | 0.090 |
| UPCR, mg/g | 598.5 (209.5, 2428) | 775.0 (233.0, 1848) | 0.884 |
| Medications used, n (%) | | | |
| Antiplatelets/Warfarin | 330 (75.0%) | 974 (44.3%) | <0.001 |
| ACEI/ARB | 337 (76.6%) | 1636 (74.4%) | 0.357 |
| B-blocker | 248 (56.4%) | 1114 (50.6%) | 0.032 |
| CCB | 285 (64.8%) | 1663 (75.6%) | <0.001 |
| Diuretics | 329 (74.8%) | 1130 (51.4%) | <0.001 |
| Statin | 183 (41.6%) | 936 (42.5%) | 0.751 |
| Oral antidiabetic agents | 216 (49.1%) | 885 (40.2%) | <0.001 |
| Insulin | 116 (26.4%) | 386 (17.5%) | <0.001 |
| NSAIDs | 135 (30.7%) | 683 (31.0%) | 0.925 |

*(Continued)*

**Table 1.** (Continued)

| Characteristics | With Digoxin (n = 440) | Without Digoxin (n = 2,200) | *P* value |
|---|---|---|---|
| **Duration of follow-up, mo, mean ± SD** | 24.8 ± 14.0 | 26.4 ± 14.8 | 0.031 |

Footnote: eGFR, Estimated GFR; SBP, systolic blood pressure; DBP, diastolic blood pressure; UPCR, urine protein to creatinine ratio; ACEI, angiotensin-converting enzyme inhibitors; ARB, angiotensin receptor blockers; CCB, calcium channel blockers; NSAIDs, nonsteroidal anti-inflammatory drugs

[a]Socioeconomic status: low economics = Dependent; moderate economics = NT$ <20000; high economics = NT$ ≥20000

[b]Comorbidity defined as once inpatient or twice outpatient records one year before index date

[c]Comorbidities score was defined as Taiwan index for hemodialysis (Reference: Clin J Am Soc Nephrol 9: 513–519, 2014)

associated with higher mortality (adjusted hazard ratio, aHR 1.63; 95% CI 1.23–2.17). However, digoxin users were not associated with higher major cardiovascular events compared to digoxin non-users (sHR 1.33, 95% CI 0.95–1.86; Table 2, model 3). The risk of single cardiovascular event was not significantly different between two groups after adjusting for covariates, including ACS (sHR 1.18, 95% CI 0.75–1.86), ischemic stroke (sHR 1.42, 95% CI 0.85–2.37), and hemorrhagic stroke (sHR 1.30, 95% CI 0.44–3.87). No difference in risk was found in renal outcomes, including ESRD (sHR 1.18, 95% CI 0.75–1.86), rapid eGFR decline (sHR 1.18, 95% CI 0.75–1.86), and AKI (aHR 1.20, 95% CI 0.87–1.64).

AT analysis revealed similar results, with significantly higher all-cause mortality in the digoxin user group compared to the non-user group (aHR 2.06, 95% CI 1.47–2.88) after

**Table 2. Association between digoxin used or not and all-cause mortality, major cardiovascular events, and renal function decline in patients with chronic kidney disease using intention-to-treat analysis.**

| Variable | Overall events | | | Adjusted Hazard Ratio (95% CI) | | |
|---|---|---|---|---|---|---|
| | With Digoxin use | Without Digoxin used | IRR (95% CI) | Model 1[a] | Model 2[b] | Model 3[c] |
| **All-cause mortality**[§] | 113 | 268 | 2.25 (1.81–2.80)*** | 1.73 (1.32–2.27)*** | 1.86 (1.41–2.45)*** | 1.63 (1.23–2.17)*** |
| **Major cardiovascular events**[¶] | 73 | 212 | 1.88 (1.44–2.46)*** | 1.59 (1.13–2.22)** | 1.70 (1.20–2.40)** | 1.33 (0.95–1.86) |
| Acute coronary syndrome[¶] | 40 | 116 | 1.86 (1.30–2.67)*** | 1.33 (0.85–2.09) | 1.41 (0.89–2.24) | 1.18 (0.75–1.86) |
| Ischemic stroke[¶] | 32 | 90 | 1.91 (1.28–2.86)** | 1.74 (1.04–2.91)* | 1.79 (1.07–3.00)* | 1.42 (0.85–2.37) |
| Hemorrhagic stroke[¶] | 5 | 23 | 1.16 (0.44–3.06) | 1.20 (0.44–3.26) | 1.35 (0.48–3.77) | 1.30 (0.44–3.87) |
| **End-stage renal disease**[¶] | 55 | 370 | 0.79 (0.60–1.05) | 0.65 (0.47–0.91)* | 0.78 (0.55–1.12) | 0.80 (0.55–1.14) |
| **Rapid eGFR decline**[¶#] | 117 | 518 | 1.14 (0.93–1.39) | 1.08 (0.85–1.37) | 1.10 (0.86–1.39) | 1.00 (0.78–1.27) |
| **Acute kidney injury**[¶] | 88 | 284 | 1.70 (1.34–2.16)*** | 1.27 (0.93–1.72) | 1.39 (1.02–1.90)* | 1.20 (0.87–1.64) |

Footnote: IRR, incidence rate ratio; eGFR, estimated GFR

[§]All-cause mortality was assessed using a Cox proportional hazard model.

[¶]Cardiovascular and renal outcomes were assessed using a Fine & Gray subdistribution hazard model for competing risk of mortality.

[#]Rapid eGFR decline defined as one year eGFR slope > 5 mL/min per 1.73 $m^2$

[a]Model 1: Adjusted for urbanization, socioeconomic status, comorbidities (diabetes mellitus, hypertension, hyperlipidemia, coronary artery disease, cerebrovascular disease, heart failure, gout, and malignancy).

[b]Model 2: Adjusted for urbanization, socioeconomic status, comorbid disorders, clinical characteristics (eGFR, systolic blood pressure, diastolic blood pressure, hematocrit, serum albumin, urine protein creatinine ratio).

[c]Model 3: Adjusted for urbanization, socioeconomic status, comorbid disorders, clinical characteristics, medications (antiplatelets, warfarin, angiotensin-converting enzyme inhibitor/angiotensin receptor blockers, calcium channel blocker, beta blocker, calcium channel blocker, diuretics, statin, oral antidiabetic agents, insulin, and nonsteroidal anti-inflammatory drugs).

* $P < 0.05$;

** $P < 0.01$;

*** $P < 0.001$

adjusting for covariates (Table 3). There was no difference in cardiovascular events or renal outcomes on AT analysis.

## Sensitivity analysis

The above results demonstrated that digoxin users were consistently associated with a higher all-cause mortality rate than digoxin non-users. Using the Cox proportional hazards model, the aHRs were 1.58 (95% CI 1.09–2.28) in the propensity score matching model with ITT analysis, 2.09 (95% CI 1.31–3.32) in the propensity score matching model with AT analysis, 1.68 (95% CI 1.24–2.28) with re-defined digoxin users as cumulative use ≥28 days, 2.53 (95% CI 1.62–3.94) with re-defined digoxin users as cumulative use ≥56 days, and 2.95 (95% CI 1.62–5.39) with re-defined digoxin users as cumulative use ≥84 days (Table 4). Similarly, cardiovascular events or renal outcomes were not significantly different between these two groups.

## Discussion

In the observational study of advanced CKD patients (stages 3b to 5) using ITT and AT analysis, digoxin use was associated with increased mortality after adjusting for patient characteristics, comorbidities, and co-administered medications. This result remained consistent in the propensity score match and when different definitions for digoxin users were used. There was no significant difference in major cardiovascular events and renal outcomes between digoxin users and non-users.

**Table 3. Association between digoxin used or not and all-cause mortality, major cardiovascular events, and renal function decline in patients with chronic kidney disease using as treat analysis.**

| Variable | Overall events | | | Adjusted Hazard Ratio (95% CI) | | |
|---|---|---|---|---|---|---|
| | With Digoxin use | Without Digoxin used | IRR (95% CI) | Model 1[a] | Model 2[b] | Model 3[c] |
| **All-cause mortality**[§] | 60 | 268 | 2.63 (1.99–3.48)*** | 2.43 (1.76–3.36)*** | 2.28 (1.64–3.16)*** | 2.06 (1.47–2.88)*** |
| **Major cardiovascular events**[¶] | 65 | 184 | 4.35 (3.28–5.77)*** | 1.80 (1.15–2.83)* | 1.82 (1.14–2.91)* | 1.49 (0.93–2.39) |
| **Acute coronary syndrome**[¶] | 34 | 96 | 4.33 (2.93–6.40)*** | 1.80 (1.01–3.22)* | 1.82 (1.00–3.29)* | 1.62 (0.89–2.94) |
| **Ischemic stroke**[¶] | 29 | 79 | 4.32 (2.83–6.62)*** | 1.59 (0.76–3.32) | 1.51 (0.72–3.20) | 1.23 (0.56–2.67) |
| **Hemorrhagic stroke**[¶] | 5 | 22 | 2.68 (1.01–7.07)* | 1.55 (0.38–6.27) | 1.68 (0.40–7.06) | 1.57 (0.32–7.66) |
| **End-stage renal disease**[¶] | 24 | 370 | 0.76 (0.50–1.15) | 0.67 (0.42–1.07) | 0.66 (0.40–1.09) | 0.67 (0.40–1.12) |
| **Rapid eGFR decline**[¶#] | 67 | 518 | 1.37 (1.06–1.76)* | 1.30 (0.98–1.72) | 1.26 (0.95–1.67) | 1.15 (0.87–1.54) |
| **Acute kidney injury**[¶] | 72 | 216 | 4.21 (3.22–5.49)*** | 1.51 (0.98–2.34) | 1.49 (0.96–2.32) | 1.30 (0.83–2.03) |

Footnote: IRR, incidence rate ratio; eGFR, estimated GFR

[§]All-cause mortality was assessed using a Cox proportional hazard model.

[¶]Cardiovascular and renal outcomes were assessed using a Fine & Gray subdistribution hazard model for competing risk of mortality.

[#]Rapid eGFR decline defined as one year eGFR slope > 5 mL/min per 1.73 m$^2$

[a]Model 1: Adjusted for urbanization, socioeconomic status, comorbidities (diabetes mellitus, hypertension, hyperlipidemia, coronary artery disease, cerebrovascular disease, heart failure, gout, and malignancy).

[b]Model 2: Adjusted for urbanization, socioeconomic status, comorbid disorders, clinical characteristics (eGFR, systolic blood pressure, diastolic blood pressure, hematocrit, serum albumin, urine protein creatinine ratio).

[c]Model 3: Adjusted for urbanization, socioeconomic status, comorbid disorders, clinical characteristics, medications (antiplatelets, warfarin, angiotensin-converting enzyme inhibitor/angiotensin receptor blockers, calcium channel blocker, beta blocker, calcium channel blocker, diuretics, statin, oral antidiabetic agents, insulin, and nonsteroidal anti-inflammatory drugs).

* $P < 0.05$;

** $P < 0.01$;

*** $P < 0.001$

**Table 4. Sensitivity analyses showing the outcomes among patients with chronic kidney disease receiving digoxin or not.**

| | Adjusted Hazard Ratio (95% CI)[a] | | | | |
|---|---|---|---|---|---|
| | Approach 1[b] | Approach 2[c] | Approach 3[d] | Approach 4[e] | Approach 5[f] |
| All-cause mortality[§] | 1.58 (1.09–2.28)* | 2.09 (1.31–3.32)** | 1.68 (1.24–2.28)*** | 2.53 (1.62–3.94)*** | 2.95 (1.62–5.39)*** |
| Major cardiovascular events[¶] | 1.73 (1.13–2.67)* | 1.82 (0.95–3.47) | 1.33 (0.93–1.91) | 2.15 (1.31–3.54)* | 1.85 (0.85–4.03) |
| Acute coronary syndrome[¶] | 1.52 (0.85–2.73) | 1.38 (0.61–3.15) | 1.19 (0.73–1.94) | 2.03 (0.98–4.19) | 1.62 (0.48–5.43) |
| Ischemic stroke[¶] | 1.51 (0.77–2.95) | 3.02 (0.80–11.4) | 1.48 (0.86–2.54) | 2.05 (0.95–4.42) | 1.88 (0.55–6.38) |
| Hemorrhagic stroke[¶] | Un-estimated | Un-estimated | 1.55 (0.48–5.04) | 1.37 (0.19–9.83) | 2.29 (0.20–26.9) |
| End-stage renal disease[¶] | 0.85 (0.54–1.32) | 0.49 (0.24–0.99)* | 0.80 (0.54–1.19) | 0.75 (0.38–1.48) | 1.13 (0.47–2.68) |
| Rapid eGFR decline[¶#] | 1.19 (0.88–1.62) | 0.95 (0.66–1.37) | 0.97 (0.75–1.26) | 0.91 (0.61–1.35) | 0.77 (0.44–1.34) |
| Acute kidney injury[¶] | 1.23 (0.82–1.85) | 1.14 (0.62–2.10) | 1.17 (0.83–1.67) | 1.40 (0.77–2.55) | 1.41 (0.55–3.60) |

[§]All-cause mortality was assessed using a Cox proportional hazard model.

[¶]Cardiovascular and renal outcomes were assessed using a Fine & Gray subdistribution hazard model for competing risk of mortality.

[#]Rapid eGFR decline defined as one year eGFR slope > 5 mL/min per 1.73 m$^2$

[a]Adjusted for urbanization, socioeconomic status, comorbidities (diabetes mellitus, hypertension, hyperlipidemia, coronary artery disease, cerebrovascular disease, heart failure, gout, and malignancy), clinical characteristics (eGFR, systolic blood pressure, diastolic blood pressure, hematocrit, serum albumin, urine protein creatinine ratio), medications (antiplatelets, warfarin, angiotensin-converting enzyme inhibitor/angiotensin receptor blockers, calcium channel blocker, beta blocker, calcium channel blocker, diuretics, statin, oral antidiabetic agents, insulin, and nonsteroidal anti-inflammatory drugs).

[b]Approach 1: propensity score-matched approach as intention-to-treatment analysis

[c]Approach 2: propensity score-matched approach as treated analysis

[d]Approach 3: Digoxin users defined as cumulative used ≥ 28 days

[e]Approach 4: Digoxin users defined as cumulative used ≥ 56 days

[f]Approach 5 Digoxin users defined as cumulative used ≥ 84 days

\* $P < 0.05$;

\*\* $P < 0.01$;

\*\*\* $P < 0.001$

## Digoxin and mortality

Few prospective studies are investigating the effect of digoxin on mortality in CKD patients. In a retrospective study from the U.S. Department of Veterans Affairs healthcare system (TREAT-AF study—The Retrospective Evaluation and Assessment of Therapies in AF), digoxin use was associated with increased risk of death in patients with AF across all stages of CKD, except for dialysis patients [23]. Digoxin use was associated with a 28% increased risk of death in another hemodialysis cohort from North America [24]. In addition to being consistent with these two studies, our results had two noteworthy features. Firstly, all patients from our study were of Chinese ethnicity, which was different from previous reports that mainly comprised Caucasian and African American participants. Thus, the association of digoxin and mortality in CKD patients might be independent of race. Secondly, the proportion of HF in digoxin users was much higher in our study compared with the TREAT-AF study (51.6% versus 21.3%). Thus, our study suggested that the association of digoxin and mortality might apply to CKD patients with AF and extend to CKD patients with HF.

## Digoxin and stroke

Digoxin has been reported to be associated with increased platelet and endothelial cell activation, which may predispose patients to thrombosis [25]. In two population-based cohort studies, digoxin was associated with an increased risk of ischemic stroke in patients with AF [26, 27]. Patients with low glomerular filtration rate (<60ml/min) has been shown to be associated

with an increased risk of stroke in a meta-analysis [28]. Based on the above studies, it was anticipated that CKD patients using digoxin would have an increased risk of ischemic stroke. However, our study showed that digoxin did not affect the risk of ischemic stroke. The discrepancy might be explained by the use of antiplatelet agents or warfarin, which might offset the possible thrombogenic effect of digoxin. In our study, the proportion of antiplatelet agent or warfarin use was higher among digoxin users (75%), compared with non-users (44.3%). As shown in Table 2, the aHR for ischemic stroke was significant in models 1 and 2, but not in model 3, after medication use was taken into consideration. In the above-mentioned study conducted by Chang et al. [26], only 23.9% of digoxin users had co-administration of warfarin, much lower compared with our study. In addition, a post-hoc analysis conducted by Gjesdal et al., with all patients receiving anticoagulation treatment, revealed no increase in thrombo-embolic events with digoxin use [29].

## Digoxin and renal function change

The toxicity of digoxin was manifested in several ways, mainly cardiac, neurological, and gastrointestinal, while direct renal toxicity has rarely been reported [30]. Consistently, in the current study, there was no significant difference in adverse renal outcomes for digoxin users and non-users; however, in post-hoc analysis conducted by Testani et al., digoxin was associated with renal function improvement in patients with HF, as compared with those taking a placebo [31].

There are several plausible explanations for the discordance. Patients from the current study were older and had worse baseline renal function, with a mean age of 73.9 ± 9.9 years and a mean eGFR of 24.8 ± 10.3ml/min per 1.73m$^2$, compared with patients in the study conducted by Testani et al. (63.4 ± 10.5 years and 70 ± 21.7ml/min per 1.73m$^2$ respectively) [31]. Old age and CKD have been associated with a lower probability of renal recovery from AKI [32, 33]. Aged kidneys have altered hemodynamics and physiological behavior in response to renal insults, which impair their ability to withstand and recover from injury [34]. In addition, the proportion of patients with HF in our study was 51.6% and 10.8% for digoxin users and non-users, respectively, while all patients in the report of Testani et al. [31] had HF. HF could cause renal dysfunction through hemodynamic changes and neurohormonal effects, termed cardiorenal syndrome [35]. Although the causes of renal dysfunction in both studies were unknown, patients with HF were more likely to have cardiorenal syndrome, which might be reversible once cardiac contractility was improved through digoxin treatment.

## Limitations

Several limitations need to be considered in our cohort study design. Firstly, the Pre-ESRD program database collected the data mainly related to renal care only. Some data were not available in this cohort database. For instance, serum digoxin concentrations (SDC) were not measured. Previous studies reported that high SDC (>1.2 ng/mL) might be associated with increased mortality when compared with low SDC (0.5–0.8 ng/mL) [36, 37]. Moreover, serum potassium levels were also unavailable in this cohort database. Hyperkalemia might decrease the effectiveness of digoxin, whereas hypokalemia could potentiate its toxicity.

Secondly, digoxin users might be frailer due to concomitant HF or AF compared to digoxin non-users. The presence of HF was defined based on ICD-9-CM coding; however, the severity of HF was unavailable due to the lack of NYHA classification and ejection fraction. Thus, full matching according to HF status in both groups was not likely, which may lead to biases.

Thirdly, the final limitation that should be considered is the very small portion of patients who received digoxin. These patients are, therefore, not fully representative of the whole

cohort. Our study aimed to enroll more digoxin users by defining digoxin users as any single digoxin exposure within three months before the index date, but only 440 digoxin users were enrolled in total. It might be explained by the stringent indication or safety concern on prescribing digoxin for CKD patients.

Despite these limitations, we try to reduce the influence of confounding factors on the outcomes by multivariable adjustment and propensity score matching and using different statistical analysis approaches. However, the inherent weaknesses of the population-based cohort study design may limit the generalizability of our findings, and we should interpret the results with caution.

In conclusion, our study demonstrated that digoxin use was associated with increased mortality in advanced CKD patients. Thus, digoxin should be prescribed with caution in this population. It warrants future prospective and randomized studies to determine the safety of digoxin in the advanced CKD patients.

## Supporting information

**S1 Table. ICD-9-CM codes used to identify clinical conditions.**
(DOCX)

**S2 Table. Drugs prescriptions during observation period among patients with chronic kidney disease.**
(DOCX)

**S1 Checklist. STROBE statement—checklist of items that should be included in reports of cohort studies.**
(DOCX)

## Acknowledgments

This study was based on data from the National Health Insurance Research Database (NHIRD) of Taiwan provided by the Bureau of National Health Insurance, Department of Health, and managed by the National Health Research Institutes. The interpretation and conclusions contained herein do not represent the views of the Bureau of National Health Insurance, Department of Health, or National Health Research Institutes.

## Author Contributions

**Conceptualization:** Shan-Min Hsu, Ping-Hsun Wu, Ming-Yen Lin, Jer-Chia Tsai.

**Data curation:** Teng-Hui Huang.

**Formal analysis:** Teng-Hui Huang.

**Funding acquisition:** Lii-Jia Yang, Jer-Chia Tsai.

**Investigation:** Teng-Hui Huang.

**Methodology:** Ping-Hsun Wu, Ming-Yen Lin, Jer-Chia Tsai.

**Supervision:** Jer-Chia Tsai.

**Writing – original draft:** Lii-Jia Yang, Shan-Min Hsu, Ping-Hsun Wu.

**Writing – review & editing:** Lii-Jia Yang, Shan-Min Hsu, Ping-Hsun Wu, Ming-Yen Lin, Teng-Hui Huang, Yi-Ting Lin, Hung-Tien Kuo, Yi-Wen Chiu, Shang-Jyh Hwang, Jer-Chia Tsai, Hung-Chun Chen.

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
