## [Decision Letter · Decision Letter 0]

8 Sep 2020

PONE-D-20-21787

Association of digoxin with mortality in patients with advanced chronic kidney disease: A population-based cohort study

PLOS ONE

Dear Dr. Tsai,

Thank you for submitting your manuscript to PLOS ONE. After careful consideration, we feel that it has merit but does not fully meet PLOS ONE’s publication criteria as it currently stands. Therefore, we invite you to submit a revised version of the manuscript that addresses the points raised during the review process.

Please note that both reviewers made critical comments on your manuscript. In particular, they identify potential bias of the results that are not adequately addressed in the discussion. They also raised some concerns about the cohort. You must adequately answer these issues. If you are not able to do so, there is a significant chance that the paper will be rejected.

We look forward to receiving your revised manuscript.

Kind regards,

Hans-Peter Brunner-La Rocca, M.D.

Academic Editor

PLOS ONE

Journal Requirements:

2. Please provide the full name of the ethics committee which approved this study in the ethics statement on the online submission form. Currently this information is only available in the methods section of your manuscript.

3. In the ethics statement in the manuscript and in the online submission form, please provide additional information about the patient records used in your retrospective study. Specifically, please ensure that you have discussed whether all data were fully anonymized before you accessed them and/or whether the IRB or ethics committee waived the requirement for informed consent. If patients provided informed written consent to have data from their medical records used in research, please include this information.

"I have read the journal's policy and the authors of this manuscript have the following competing interests: Co-author Ping-Hsun Wu is a fellow of PLOS ONE Editorial Board Members. This does not alter the authors’ adherence to PLOS ONE editorial policies and criteria."

Reviewers' comments:

Reviewer's Responses to Questions

**Comments to the Author**

1. Is the manuscript technically sound, and do the data support the conclusions?

Reviewer #1: Yes

Reviewer #2: Partly

2. Has the statistical analysis been performed appropriately and rigorously? 

Reviewer #1: I Don't Know

Reviewer #2: No

3. Have the authors made all data underlying the findings in their manuscript fully available?

Reviewer #1: Yes

Reviewer #2: Yes

4. Is the manuscript presented in an intelligible fashion and written in standard English?

Reviewer #1: Yes

Reviewer #2: Yes

5. Review Comments to the Author

Reviewer #1: Overall comments:

Yang and colleagues use a national dataset of Taiwanese patients with chronic kidney disease stage 3b or worse (eGFR <45 mL/min/1.73m2) or with at least 1g proteinuria, to assess the association between digoxin use and the risk of death, cardiovascular disease and renal outcomes. They found that those taking digoxin were at higher risk of death, compared to those not taking it, but the risk of cardiovascular event and rapid eGFR decline was not different between the groups.

Digoxin is used to treat patients with heart failure with reduced EF, typically with NYHA III-IV, and EF <25%. These patients tend to be sicker and have more comorbidities than those who do not require a treatment with digoxin (as shown in table 1 of the current study). Given the characteristics associated with the population taking digoxin, it is expected that digoxin use be associated with higher mortality risk and it is expected to remain the case in a cohort of CKD patients.

In addition, people with advanced CKD are more likely to have hyperkalemia (unfortunately data not available) than the rest of the population, which adds a potential risk to digoxin use.

Major comments:

- It would be helpful if the authors could explain in more details their statistical plan, and the choice of the various tests they use.

- My understanding is that the all cause mortality outcome was assessed using a Cox proportional hazard model while the other outcomes (cardiovascular and renal) were examined using a Fine&Gray subdistribution hazard model, taking into account the competing risk of death. This should be made more clear in the tables. The tables are currently presented as if the statistical test looking at “all cause mortality” as the outcome, is also being adjusted for the competing risk of death, which is very confusing.

- How did the authors adjust for medications ? Was there a code Y/N for each medication class, or were all the medications considered as one variable ?

- “Poisson regression model was used to analyze the incidence rate ratios (IRRs) and 95% confidence intervals (95% CIs) of outcomes (all-cause mortality, cardiovascular events, and renal outcomes) were examined for digoxin users and non-users.” This sentence is not clear. Could the authors please explain why they chose to use a Poisson regression?

Minor comments:

- Using ICD codes to define CKD and AKI events is disputable, and using an average serum creatinine value, and the KIDGO definition of AKI may have been more accurate.

- “Baseline eGFR was calculated from the last recorded serum creatinine level before the index date” how long before the index date could that serum creatinine be measured ? Could an average of 2 or 3 eGFRs be used instead ? One single value for a serum creatinine (or eGFR) can be highly misleading.

Reviewer #2: This study investigates the potential impact of digoxin use in patients with CKD. The authors identified 2640 patients in their program and found that patients taking digoxin were at higher risk of dying whereas the risk of cv events and renal function decline did not differ. The potential impact of digoxin in CKD patients is of clinical relevance.

Some comments:

The most important shortcoming of this analysis is the fact that matching was based on age and sex only. Although they adjusted for multiple comorbidities, matching for the most relevant factors would be a better option. In addition and most importantly, severity of HF is not properly defined. It is VERY LIKELY that more advanced HF patients were more likely to receive digoxin as recommended by the guidelines. This could explain the lower blood pressure in those taking digoxin. The authors must be very clear on this. The way how they discuss this issues is not sufficient. Also, the authors do not provide any evidence for their discussion about potential survivors of digoxin being included in the study. They do not have any information on this (in fact if anything can be taken from their data, it would be the opposite as longer-term treatment was numerically associated with higher risk). The authors provide a reference that new digoxin users might have a higher mortality as compared to chronic users, but this was in AF only.

The authors should stress even more that SDC were not available. This is crucial as effects on mortality highly depends on SDC (as cited by the authors).A substitute could be the dosage of digoxin in relation to renal function and body weight. If dosage is available the authors should provide such calculation. If not, they need to report this as additional shortcoming.

They authors identified almost 32,000 patients but only less than 10% were actually included in the analysis. The authors report that this reduction happened after the matching process, which considered age and sex only. Sex and age is presumably known for all 32,000 patients. Why were only so few included? Was the reason that only 440 patients received digoxin (i.e. 1.4% of the total population)? The authors should include more patients or must report as a limitation that only a very small minority received digoxin which may limit generalisability of the findings. If it is possible to include more patients, it would be interesting to see results separately for patients with HF and AF.

Where there also patients without HF and AF but receiving digoxin? If yes, what was the indication in those? Generally speaking, the authors should provide information on the indication for the use of digoxin in their cohort.

The authors refer to previous studies how ACS, ischemic stroke and haemorrhagic stroke were identified. However, the authors should briefly describe this as not all readers may have access to the referred studies. In addition, they do not report on how death was identified and verified.

The authors completed the follow-up already in 2012. What is the reason for this?

The authors should report in their conclusion that full adjustment was not possible, making bias of their findings likely.

6. PLOS authors have the option to publish the peer review history of their article (what does this mean?). If published, this will include your full peer review and any attached files.

Reviewer #1: No

Reviewer #2: No

---

## [Author Response · Author response to Decision Letter 0]

2 Nov 2020

Response to editor　

Thank the editor for the important comments. We have revised our manuscript and responded the questions as requested. Please see the response for eache question. 

1. Please ensure that your manuscript meets PLOS ONE’s style requirements, including those for file naming.

Response to Q1: 

Thank you for your comment. We have revised manuscript and file naming to meet PLOS ONE’s style requirements.

2. Please provide the full name of the ethics committee which approved this study in the ethics statement on the online submission form. Currently this information is only available in the methods section of your manuscript.

Response to Q2: 

We have provided this information on the methods section and online submission form as “This study was approved by the Institutional Review Board (IRB) of Kaohsiung Medical University Hospital (KMUHIRB-EXEMPT(I)-20180035), and the requirement for informed consent was waived.” (page 8, line 112-115). 

3. In the ethics statement in the manuscript and in the online submission form, please provide additional information about the patient records used in your retrospective study. Specifically, please ensure that you have discussed whether all data were fully anonymized before you accessed them and/or whether the IRB or ethics committee waived the requirement for informed consent. If patients provided informed written consent to have data from their medical records used in research, please include this information.

Response to Q3: 

We provided this information about the patient records in the revised manuscript (page 7, line 96-101) and online submission as “We conducted a retrospective cohort study using the Pre-ESRD care program registry linked with the National Health Insurance Research Database (NHIRD), containing detailed information on inpatient and outpatient services. To protect patients’ privacy, NHIRD had made all data fully anonymized by replacing all personal identification with surrogate numbers before researchers accessed them and further analyzed them.” 

Moreover, our local IRB ethics committee has waived the requirement for informed consent. This statement is stated in the revised manuscript (page 8, line 112-115) as “This study was approved by the Institutional Review Board (IRB) of Kaohsiung Medical University Hospital (KMUHIRB-EXEMPT(I)-20180035), and the requirement for informed consent was waived.”

“I have read the journal’s policy and the authors of this manuscript have the following competing interests: Co-author Ping-Hsun Wu is a fellow of PLOS ONE Editorial Board Members. This does not alter the authors’ adherence to PLOS ONE editorial policies and criteria.” Please confirm that this does not alter your adherence to all PLOS ONE policies on sharing data and materials, by including the following statement: “This does not alter our adherence to PLOS ONE policies on sharing data and materials.” 

If there are restrictions on sharing of data and/or materials, please state these. Please note that we cannot proceed with consideration of your article until this information has been declared.

Please know it is PLOS ONE policy for corresponding authors to declare, on behalf of all authors, all potential competing interests for the purposes of transparency. PLOS defines a competing interest as anything that interferes with, or could reasonably be perceived as interfering with, the full and objective presentation, peer review, editorial decision-making, or publication of research or non-research articles submitted to one of the journals. Competing interests can be financial or non-financial, professional, or personal. Competing interests can arise in relationship to an organization or another person. 

Response to Q4: 

We have confirmed the competing interests in the following statement in the revived manuscript (page 21, line 359-360) as:

“Co-author Ping-Hsun Wu is a fellow of PLOS ONE Editorial Board Members. This does not alter our adherence to PLOS ONE policies on sharing data and materials.” 

5. We note that you have indicated that data from this study are available upon request. PLOS only allows data to be available upon request if there are legal or ethical restrictions on sharing data publicly.

b) If there are no restrictions, please upload the minimal anonymized data set necessary to replicate your study findings as either Supporting Information files or to a stable, public repository and provide us with the relevant URLs, DOIs, or accession numbers. Please see http://www.bmj.com/content/340/bmj.c181.long for guidelines on how to de-identify and prepare clinical data for publication. 

Response to Q5: 

The legal and ethical restrictions in Taiwan are the main reasons for not allowing to share data publicly. We stated this explanation in the revised manuscript (page 20, line 344-357) as below. “The raw data used in our study were obtained from NHIRD and "pre-ESRD care program dataset" through formal application. The datasets we used from NHIRD included "H_NHI_OPDTE, H_NHI_IPDTE, H_NHI_DRUGE, H_NHI_OPDTO, H_NHI_IPDTO, H_NHI_DRUGO, H_NHI_ENROL, H_NHI_CATAS, and H_OST_DEATH". Data holder for NHIRD was the Health and Welfare Data Science Center, Department of Statistics, Ministry of Health and Welfare, Taiwan (https://dep.mohw.gov.tw/DOS/np-2497-113.html). Data holder for "Pre-ESRD care program dataset" was Information Integration and Application Center, National Health Insurance Administration, Ministry of Health and Welfare (https://www.nhi.gov.tw/Content_List.aspx?n=2D2FAF5214807829&topn=787128DAD5F71B1A). These raw data were limited to research purposes only and cannot be made publicly available under regulation of the "Personal Information Protection Act" in Taiwan. We, as authors, did not have any special access privileges to the data that other researchers would not have.” 

 

Response to Reviewer 1 

Major comments:

Q1: It would be helpful if the authors could explain in more details their statistical 

plan, and the choice of the various tests they use. My understanding is that the all cause mortality outcome was assessed using a Cox proportional hazard model while the other outcomes (cardiovascular and renal) were examined using a Fine&Gray subdistribution hazard model, taking into account the competing risk of death. This should be made more clear in the tables. The tables are currently presented as if the statistical test looking at “all cause mortality” as the outcome, is also being adjusted for the competing risk of death, which is very confusing.

Response to Q1: 

Thank you for the comment. Regarding the statistical plan and the choice of the various tests, we provide this information in the revised manuscript (page 11, line 161-164 ) as “Regarding the all-cause mortality outcome, we employed Cox proportional hazards model. Furthermore, we applied the Fine and Gray subdistribution hazards model to clarify the competing risk of death and the effects of digoxin on the cardiovascular and renal outcomes [21].”

We also add this information in the footnote of Table 2 to 4 as “All-cause mortality was assessed using a Cox proportional hazard model. Effects of digoxin on the cardiovascular and renal outcomes were adjusted for competing risk of mortality using a Fine & Gray subdistribution hazard model.”

Q2: How did the authors adjust for medications? Was there a code Y/N for each medication class, or were all the medications considered as one variable?

Response to Q2:

We defined each medication class as one variable with a Y/N code. We revised the sentence in page11, line 164-166, as “We applied the multivariable models to adjust the confounders for urbanization, socioeconomic status, comorbid disorders, clinical characteristics, and each medication class.” 

Q3: “Poisson regression model was used to analyze the incidence rate ratios (IRRs) and 95% confidence intervals (95% CIs) of outcomes (all-cause mortality, cardiovascular events, and renal outcomes) were examined for digoxin users and non-users.” This sentence is not clear. Could the authors please explain why they chose to use a Poisson regression?

Response to Q3: 

Poisson regression model can be applied to evaluate relative risk of outcomes between groups and complex interactions with covariates [1]. Compared with Cox proportional hazards model, Poisson regression model was more appropriate to evaluate relative risk of outcomes between our groups across all periods [2]. Thus, we used the Poisson regression model to estimate the effect of digoxin on the incidence rates of outcomes in this study. We revised the sentence in the revised manuscript (page 10, line 158-160) as “The incidence rate ratios (IRRs) and 95% confidence intervals (95% CIs) of outcomes (all-cause mortality, cardiovascular events, and renal outcomes) for digoxin users versus non-users were examined by using the Poisson regression model [20].” 

Minor comments:

Q4: Using ICD codes to define CKD and AKI events is disputable, and using an average serum creatinine value, and the KIDGO definition of AKI may have been more accurate.

Response to Q4:

For the question about using ICD codes to define CKD and AKI events, we agree that the KIDGO definition of AKI may have been more accurate. However, previous reports also supported the rationale of using ICD codes to define AKI events. According to the study conducted by Waikar et al., ICD-9-CM codes 584 for AKI diagnosis had a sensitivity of 35.4%, specificity of 97.7%, the positive predictive value of 47.9%, and negative predictive value of 96.1% [3]. Based on this study, using the ICD-9-CM code 584 is reliable for AKI events because of high specificity although relatively low sensitivity may miss some potential candidates.

Next, for the question about using ICD codes to define CKD, we selected the CKD population for this study from pre-ESRD program registry database in addition to ICD-CM codes 585 and 581.9. The eligibility criteria for pre-ESRD program were individuals with CKD stages 3b-5 (eGFR<45 mL/min/1.73 m2), or those with proteinuria (urine protein to creatinine ratio, UPCR >1000 mg/g). The duration of the illness must be more than 3 months before enrollment to the pre-ESRD program. Thus, patients identified through these processes fit the definition of CKD by KDIGO. 

Q5: “Baseline eGFR was calculated from the last recorded serum creatinine level before the index date” how long before the index date could that serum creatinine be measured ? Could an average of 2 or 3 eGFRs be used instead ? One single value for a serum creatinine (or eGFR) can be highly misleading.

Response to Q5:

For this question, we employed the enrollment criteria for entering the pre-ESRD program to explain that baseline eGFR was calculated by last recorded serum creatinine levels. The Enrollment criteria for entering the pre-ESRD program included patients with CKD stage 3b-5 or those with proteinuria >1g/day. All patients were followed up at least quarterly. The consensus for enrollment criteria was stringent for the nephrologists to select the candidate patients under NHI regulation. Thus, we considered that a single value of serum creatinine could be used as the measurement for baseline eGFR data. 

 

Response to Reviewer 2 

Q1: The most important shortcoming of this analysis is the fact that matching was based on age and sex only. Although they adjusted for multiple comorbidities, matching for the most relevant factors would be a better option. 

Response to Q1:

We agree that adding more relevant factors in matching would make two groups more comparable in addition to age and sex matching. However, this way might reduce the sample sizes and overall statistical power. To ensure having enough patients for analysis and maintain sufficient statistical power, we aimed to match patients by age and sex for the primary analysis. Furthermore, we applied propensity scores matching to consider all covariates to increase comparability and reduce potential confounding effects between two groups. Both matching approaches consistently showed that digoxin could exert a higher risk for mortality of the study population.

Q2: In addition and most importantly, severity of HF is not properly defined. It is VERY LIKELY that more advanced HF patients were more likely to receive digoxin as recommended by the guidelines. This could explain the lower blood pressure in those taking digoxin. The authors must be very clear on this. The way how they discuss this issues is not sufficient. 

Response to Q2:

In our cohort study design, the Pre-ESRD program database collected the data mainly related to renal care. Severity of HF was not available in this database. However, previous study showed that diuretics use could be a proxy for reflecting the severity of HF [4]. Thus, we adjusted confounding effect of severity of HF by the variable of diuretics use. In this way, the influence of the severity of HF on the outcomes could be attenuated after adjusting diuretics use. We admit that we cannot collect and well control all confounders due to inherent limitations of cohort study design. We have addressed these limitations in the Discussion. 

 The revision paragraph was shown in the limitations section of Discussion (page 

18, line 306-308) as “Furthermore, cardiac functional status by NYHA classification and structural parameters by ejection fraction were unavailable for us to evaluate their impacts on the outcomes.”

Q3: Also, the authors do not provide any evidence for their Discussion about potential survivors of digoxin being included in the study. They do not have any information on this (in fact if anything can be taken from their data, it would be the opposite as longer-term treatment was numerically associated with higher risk). The authors provide a reference that new digoxin users might have a higher mortality as compared to chronic users, but this was in AF only.

Response to Q3:

 We admit that this study lacked the information about digoxin chronic users or new users due to limitations of the cohort database. We agree with your concern that our previous discussion about this issue might not be convincing. Thus, we decide to delete the following paragraph in the discussion as “On the other hand, chronic digoxin users might not be as vulnerable because they might have already survived the potential harm imposed by this drug. Otherwise, they would have died, or digoxin would have been discontinued. It has been shown that in AF patients, new digoxin users were associated with higher mortality compared with non-users, while chronic digoxin users were not” (p17 in the previous manuscript). 

Q4: The authors should stress even more that SDC were not available. This is crucial as effects on mortality highly depends on SDC (as cited by the authors).A substitute could be the dosage of digoxin in relation to renal function and body weight. If dosage is available the authors should provide such calculation. If not, they need to report this as additional shortcoming.

Response to Q4:

It was reported that serum digoxin concentration (SDC) could be affected by many factors, such as renal function, the bioavailability of the digoxin formulation used, the volume of distribution, the amount of extrarenal clearance, body weight, and serum albumin concentration [5]. Although some equations have been proposed to predict the SDC, their validity in Asians or advanced CKD patients was still limited [6] [7]. Thus, there might be concerns to calculate SDC by these equations in our study population. We have stated the lack of SDC as a limitation in the Discussion (Page 18, line 301-304) as “For instance, serum digoxin concentrations (SDC) were noted measured. Previous studies reported that high SDC (>1.2 ng/mL) might be associated with increased mortality when compared with low SDC (0.5–0.8 ng/mL) [36,37].” 

Q5: They authors identified almost 32,000 patients but only less than 10% were actually included in the analysis. The authors report that this reduction happened after the matching process, which considered age and sex only. Sex and age is presumably known for all 32,000 patients. Why were only so few included? Was the reason that only 440 patients received digoxin (i.e. 1.4% of the total population)? The authors should include more patients or must report as a limitation that only a very small minority received digoxin which may limit generalisability of the findings. If it is possible to include more patients, it would be interesting to see results separately for patients with HF and AF.

Response to Q5:

The small sample size could be explained by that only a small portion of CKD patients met the indication for digoxin, rather than by matching process. Two reasons may help explain this phenomenon. First, guidelines do not recommend digoxin as the first-line medication for HF, and digoxin was used mainly for symptomatic HF patients despite receiving standard therapy. Second, the information about the safety of digoxin in advanced CKD patients was limited. We stated this issue in the limitation section in the Discussion (page 18, line 309-313) as below. “Another limitation was the relatively small sample sizes of digoxin users. Our study aimed to enroll more digoxin users by defining digoxin users as any single digoxin exposure within three months before the index date, but only 440 digoxin users were enrolled in total. It might be explained by the stringent indication or safety concern on prescribing digoxin for CKD patients.” 

Q6: Where there also patients without HF and AF but receiving digoxin? If yes, what was the indication in those? Generally speaking, the authors should provide information on the indication for the use of digoxin in their cohort.

Response to Q6:

To our knowledge, AF and HF were the main indications for digoxin use. However, the indication for digoxin in each patient was not well recorded in the National Health Insurance (NHI) claim database. Some reasons related to the NHI claim database’s regulations may explain why patients without HF and AF might receive digoxin. First, only the first three diagnosis codes were recorded in the National Health Insurance claim database. Thus, the corresponding ICD codes for AF or HF might be missed because they were placed in the latter part of the diagnosis list in the medical record. Second, digoxin might be prescribed for the off-label condition, such as supraventricular tachycardia, not for HF or AF. 

Q7: The authors refer to previous studies how ACS, ischemic stroke and haemorrhagic stroke were identified. However, the authors should briefly describe this as not all readers may have access to the referred studies. 

Response to Q7:

We defined and identified the ACS, ischemic stroke, and hemorrhagic stroke based on the ICD-9-CM codes. The detailed information for this issue was stated in the Method section (page 9, line 121-126) as below. “ACS, ischemic stroke, and hemorrhagic stroke were defined as hospitalization for these vascular events, which were validated in previous studies [15,16]. For example, ICD-9-CM codes 433 (occlusion of cerebral arteries) and 434 (stenosis of precerebral arteries) were used to extract from NHIRD study subjects with ischemic stroke and were admitted for the specific diagnosis.”

Q8: In addition, they do not report on how death was identified and verified.

Response to Q8:

We identified and verified the outcome of death based on the evidence of patient withdrawal from the NHI claim database. It was reported to be valid to identify the death outcome using a similar approach [8]. We stated this issue in the Method section (page 8, line 120-121) as “Death was ascertained based on the evidence of patient withdrawal from the NHI claim database.” 

Q9: The authors completed the follow-up already in 2012. What is the reason for this? 

Response to Q9:

We state the following explanations for this question. First, there was a time lag between reporting the claims to the NHI database and its release for research purposes. At that time of our application for data analysis, the patient registry time was the year 2012. Second, it takes time to process and analyze the enormous cohort database and to validate their accuracy. We aimed to ensure the robustness of research design and comprehensiveness of statistical analysis for our work. 

Q10: The authors should report in their conclusion that full adjustment was not possible, making bias of their findings likely.

Response to Q10:

We are aware that full adjustment was not possible, as you mentioned. We stated this limitation in the Discussion (page 19, lime 316-320) as below. “Despite these limitations, we try to reduce the influence of confounding factors on the outcomes by multivariable adjustment and propensity score matching and using different statistical analysis approaches. The inherent weaknesses of the population-based cohort study design may limit the generalizability of our findings. We should interpret the results with caution.”

References 

1. Frome, E.L. and H. Checkoway, Epidemiologic programs for computers and calculators. Use of Poisson regression models in estimating incidence rates and ratios. Am J Epidemiol, 1985. 121(2): p. 309-23.

2. Vonesh EF, Schaubel DE, Hao W, Collins AJ. Statistical methods for comparing mortality among ESRD patients: Examples of regional/international variations. Kidney International, Supplement. 2000 Dec 1;57(74).

3. Waikar, S.S., et al., Validity of International Classification of Diseases, Ninth Revision, Clinical Modification Codes for Acute Renal Failure. J Am Soc Nephrol, 2006. 17(6): p. 1688-94.

4. Gislason GH, et al. Persistent use of evidence-based pharmacotherapy in heart failure is associated with improved outcomes. Circulation. 2007; 116(7):737-44.

5. Iisalo, E., Clinical pharmacokinetics of digoxin. Clin Pharmacokinet, 1977. 2(1): p. 1-16.

6. Zhao, L., et al., Efficiency of individual dosage of digoxin with calculated concentration. Clin Interv Aging, 2014. 9: p. 1205-10.

7. Muzzarelli, S., et al., Individual dosage of digoxin in patients with heart failure. Qjm, 2011. 104(4): p. 309-17.

8. Cheng C-L, Chien H-C, Lee C-H, Lin S-J, Yang Y-HK. Validity of in-hospital mortality data among patients with acute myocardial infarction or stroke in National Health Insurance Research Database in Taiwan. International Journal of Cardiology. 2015; 201:96-101.

---

## [Decision Letter · Decision Letter 1]

27 Nov 2020

PONE-D-20-21787R1

Association of digoxin with mortality in patients with advanced chronic kidney disease: A population-based cohort study

PLOS ONE

Dear Dr. Tsai,

Thank you for submitting your manuscript to PLOS ONE. After careful consideration, we feel that it has merit but does not fully meet PLOS ONE’s publication criteria as it currently stands. Therefore, we invite you to submit a revised version of the manuscript that addresses the points raised during the review process.

Please note that the reviewers still have some minor issues to be resolved. In particular, reviewer #2 asks for clarity in your statement about the interpretation of the findings. The conclusion that caution is required when giving digoxin to these patients, it must be also mentioned what is lacking and not trying to circumvent clear statements. Please have a second look also at the initial comments by reviewer #2.

We look forward to receiving your revised manuscript.

Kind regards,

Hans-Peter Brunner-La Rocca, M.D.

Academic Editor

PLOS ONE

Reviewers' comments:

Reviewer's Responses to Questions

**Comments to the Author**

1. If the authors have adequately addressed your comments raised in a previous round of review and you feel that this manuscript is now acceptable for publication, you may indicate that here to bypass the “Comments to the Author” section, enter your conflict of interest statement in the “Confidential to Editor” section, and submit your "Accept" recommendation.

Reviewer #1: All comments have been addressed

Reviewer #2: (No Response)

2. Is the manuscript technically sound, and do the data support the conclusions?

Reviewer #1: Yes

Reviewer #2: Partly

3. Has the statistical analysis been performed appropriately and rigorously? 

Reviewer #1: Yes

Reviewer #2: Yes

4. Have the authors made all data underlying the findings in their manuscript fully available?

Reviewer #1: Yes

Reviewer #2: Yes

5. Is the manuscript presented in an intelligible fashion and written in standard English?

Reviewer #1: Yes

Reviewer #2: Yes

6. Review Comments to the Author

Reviewer #1: The authors have answered all my comments. I still believe however, that the tables lack clarity. The outcome "all-cause mortality" is analyzed with various models which all adjust for the competing risk of death (as stated in the foot note). An asterisk should be noted there, with mention that, for this outcome, models adjust for X, Y, Z,... but not for competing risk of death.

Reviewer #2: The authors have improved the manuscript. However, they still lack sufficiently clear statements that important clinical data are missing. It is not only NYHA-class and ejection fraction, but they lack information about the presence of heart failure. This must be very clear in the text. This also means that full propensity score matching is not possible. They should also state in the conclusion that full matching was not possible to make very clear that the interpretation of their data must be done with caution.

They should also mention as a limitation not specifically the relatively small sample size, but the fact that only a very small portion of patients received digoxine. These patients are therefore likely not representative for the whole cohort.

7. PLOS authors have the option to publish the peer review history of their article (what does this mean?). If published, this will include your full peer review and any attached files.

Reviewer #1: No

Reviewer #2: No

---

## [Author Response · Author response to Decision Letter 1]

4 Dec 2020

Dear Editor and reviewers,

 We appreciate your relevant comments from the Editor and reviewers. We have revised the manuscript based on reviewers’ comments and suggestions as below. Please kindly check the updated submission of the revised manuscript.

Kind regards

Jer-Chia Tsai, MD. 

Response to Reviewer 1 

Q1: The authors have answered all my comments. I still believe however, that the tables lack clarity. The outcome “all-cause mortality” is analyzed with various models which all adjust for the competing risk of death (as stated in the foot note). An asterisk should be noted there, with mention that, for this outcome, models adjust for X, Y, Z,... but not for competing risk of death.

Response to Q1:

Thank you for the comment. To clarify the statistical analysis for each outcome in Table 2, 3, and 4, we labeled the marks and the corresponding statistical analysis as:

“§All-cause mortality was assessed using a Cox proportional hazard model, and ¶Cardiovascular and renal outcomes were assessed using a Fine & Gray subdistribution hazard model for competing risk of mortality.” 

 

Response to Reviewer 2 

Q1: The authors have improved the manuscript. However, they still lack sufficiently clear statements that important clinical data are missing. It is not only NYHA-class and ejection fraction, but they lack information about the presence of heart failure. This must be very clear in the text. This also means that full propensity score matching is not possible. They should also state in the conclusion that full matching was not possible to make very clear that the interpretation of their data must be done with caution.

Response to Q1:

Thank you for the comment. For the information about the presence of heart failure (HF), Table 1 shows the rates of HF in the digoxin user and non-user groups were 51.6% and 10.8%, respectively. The definition of HF is based on ICD-9-CM codes 398, 402, 404, and 428. We admit that this study lacks the clinical data about NYHA-class and ejection fraction. 

Following your suggestion, we state this limitation in the revision as: “Secondly, digoxin users might be frailer due to concomitant HF or AF compared to digoxin non-users. The presence of HF was defined based on ICD-9-CM coding; however, the severity of HF was unavailable due to the lack of NYHA classification and ejection fraction. Thus, full matching according to HF status in both groups was not likely, which may lead to biases.” (page 18, line 305-309) 

Furthermore, we also state the limitation of this population-based cohort study design as: “However, the inherent weaknesses of the population-based cohort study design may limit the generalizability of our findings, and we should interpret the results with caution.” (page 19, line 318-320). 

Q2: They should also mention as a limitation not specifically the relatively small sample size, but the fact that only a very small portion of patients received digoxine. These patients are therefore likely not representative for the whole cohort.

Response to Q2:

Thank you for the comment. Following your suggestion, we revise the paragraph as: “Thirdly, the final limitation should be considered is that a very small portion of patients received digoxin. These patients are, therefore, not fully representative of the whole cohort. This study aimed to enroll more digoxin users by defining digoxin users as any single digoxin exposure within three months before the index date, but only 440 digoxin users were enrolled in total. It might be explained by the stringent indication or safety concern on prescribing digoxin for CKD patients.” (page 18, line 310 to page 19, line 315)

---

## [Editor Report · Decision Letter 2]

14 Dec 2020

PONE-D-20-21787R2

Association of digoxin with mortality in patients with advanced chronic kidney disease: A population-based cohort study

PLOS ONE

Dear Dr. Tsai,

Thank you for submitting your manuscript to PLOS ONE. After careful consideration, we feel that it has merit but does not fully meet PLOS ONE’s publication criteria as it currently stands. Therefore, we invite you to submit a revised version of the manuscript that addresses the points raised during the review process.

As most of the remaining issues have been addressed, I do not send your manuscript back to the reviewers. However, I would like you to address the following points: As PLOS ONE does not provide proofreading, even small details need to be adequately addressed. In your revised manuscript you write: "Thirdly, the final limitation should be considered is that a very small portion of patients received digoxin." This is not best English and should be e.g. "Thirdly, the final limitation that should be considered is the very small portion of patients who received digoxin." In addition, you must temper your conclusions as addressed by reviewer #2. In the limitation section, you mention that the results must be interpreted with caution, but in the conclusion, you present your findings as "consistent evidence". This is simply not possible with the design of the study. You really need to temper this significantly and mainly say that digoxin should be given with caution, based on your results. You then can highlight the need for prospective testing.

We look forward to receiving your revised manuscript.

Kind regards,

Hans-Peter Brunner-La Rocca, M.D.

Academic Editor

PLOS ONE

---

## [Author Response · Author response to Decision Letter 2]

30 Dec 2020

Q1: As PLOS ONE does not provide proofreading, even small details need to be adequately addressed. In your revised manuscript you write: "Thirdly, the final limitation should be considered is that a very small portion of patients received digoxin." This is not best English and should be e.g. "Thirdly, the final limitation that should be considered is the very small portion of patients who received digoxin."

Response to Q1:

Thank you for the comment. We have revised the sentence as “Thirdly, the final limitation that should be considered is the very small portion of patients who received digoxin.” (Page 18, line 310-311)

Q2: In addition, you must temper your conclusions as addressed by reviewer #2. In the limitation section, you mention that the results must be interpreted with caution, but in the conclusion, you present your findings as "consistent evidence". This is simply not possible with the design of the study. You really need to temper this significantly and mainly say that digoxin should be given with caution, based on your results. You then can highlight the need for prospective testing.

Response to Q2:

Thank you for the comment. We have revised the paragraph as “In conclusion, our study demonstrated that digoxin use was associated with increased mortality in advanced CKD patients. Thus, digoxin should be prescribed with caution in this population. It warrants future prospective and randomized studies to determine the safety of digoxin in the advanced CKD patients.” (page 19, line 321-324)

---

## [Editor Report · Decision Letter 3]

5 Jan 2021

Association of digoxin with mortality in patients with advanced chronic kidney disease: A population-based cohort study

PONE-D-20-21787R3

Dear Dr. Tsai,

We’re pleased to inform you that your manuscript has been judged scientifically suitable for publication and will be formally accepted for publication once it meets all outstanding technical requirements.

Kind regards,

Hans-Peter Brunner-La Rocca, M.D.

Academic Editor

PLOS ONE
---

## [Editor Report · Acceptance letter]

7 Jan 2021

PONE-D-20-21787R3 

Association of digoxin with mortality in patients with advanced chronic kidney disease: A population-based cohort study 

Dear Dr. Tsai:

I'm pleased to inform you that your manuscript has been deemed suitable for publication in PLOS ONE. Congratulations! Your manuscript is now with our production department. 

Kind regards, 

on behalf of

Dr. Hans-Peter Brunner-La Rocca 

Academic Editor

PLOS ONE